# The Most Vulnerable Hispanic Immigrants in New York City: Structural Racism and Gendered Differences in COVID-19 Deaths

**DOI:** 10.3390/ijerph20105838

**Published:** 2023-05-16

**Authors:** Norma Fuentes-Mayorga, Alfredo Cuecuecha Mendoza

**Affiliations:** 1The Colin Powell School for Civic and Global Leadership, The City College New York (CCNY), New York, NY 10031, USA; 2Centro de Investigación e Inteligencia Económicas-UPAEP, Puebla 72410, Mexico; alfredo.cuecuecha@upaep.mx

**Keywords:** New York City, Hispanic population, COVID-19 death rates, Hispanic immigrants, structural racism, spatial concentration, gendered analysis, Hispanic health paradox

## Abstract

This paper explores the structural and group-specific factors explaining the excess death rates experienced by the Hispanic population in New York City during the peak years of the coronavirus pandemic. Neighborhood-level analysis of Census data allows an exploration of the relation between Hispanic COVID-19 deaths and spatial concentration, conceived in this study as a proxy for structural racism. This analysis also provides a more detailed exploration of the role of gender in understanding the effects of spatial segregation among different Hispanic subgroups, as gender has emerged as a significant variable in explaining the structural and social effects of COVID-19. Our results show a positive correlation between COVID-19 death rates and the share of Hispanic neighborhood residents. However, for men, this correlation cannot be explained by the characteristics of the neighborhood, as it is for women. In sum, we find: (a) differences in mortality risks between Hispanic men and women; (b) that weathering effects increase mortality risks the longer Hispanic immigrant groups reside in the U.S.; (c) that Hispanic males experience greater contagion and mortality risks associated with the workplace; and (d) we find evidence corroborating the importance of access to health insurance and citizenship status in reducing mortality risks. The findings propose revisiting the Hispanic health paradox with the use of structural racism and gendered frameworks.

## 1. Introduction

The U.S. COVID-19 pandemic unraveled an unprecedented health and economic crisis in which historically racialized and vulnerable minorities paid the highest price with their lives. Federal policy defines “Hispanic” not as a race but as an ethnicity. Additionally, it prescribes that Hispanics can, in fact, be of any race. However, some authors argue that standard U.S. racial categories might either be confusing or not provide relevant options for Hispanics to describe their racial identity. A good share of Hispanics considers their Hispanic background to be part of their racial background, while others think of it as their ethnic background. A good share considers the label both an ethnic and racial identity [1]. The disproportionally higher death rates decimated the Hispanic population of New York City which made it clear that legacies of structural and social inequalities combined to increase the higher risk of mortality for the group [2,3,4,5,6,7,8,9,10,11,12,13,14,15,16,17,18,19,20,21,22,23,24,25,26,27,28,29,30,31,32,33,34,35,36,37,38,39,40,41,42,43,44,45,46,47,48,49,50,51,52,53,54,55,56,57,58,59,60]. In 2020, for example, during the peak year of the pandemic, the Hispanic population experienced a decline in life expectancy at birth, shrinking the group’s survival advantage over the white population from over three years to less than one year [3]. As of June 2022, in New York City, at the epicenter of the pandemic, Hispanics exhibited the highest cumulative COVID-19 infection rates (28,025 per 100,000 people) and death rates (871 per 100,000) and the second-highest hospitalization rates (2150 per 100,000 people) among all racial and ethnic groups for whom the city collects data [56]. This is the case even when both Hispanic adults and eligible children in New York City have higher vaccination rates compared to the white and black populations [57]. Currently, at 60.6 million, Hispanics are the largest minority group and the youngest of all racial and ethnic groups, with an average age of 30 in the U.S. These young adults compose disproportionately the largest share of voters, workers, and consumers among Latinos. During the first and peak year of the pandemic, they suffered the highest excess mortality rates of any other racial or ethnic group in the nation [58]. In California, for example, Hispanics ages 20 to 54 were 8.5 times more likely than white Americans to die of COVID-19 [45]. In New Jersey, young Latino men between the ages of 18–49 died at 4.5 times the rate of Latino women and 7 times the rate of white men in the same age category. However, to this date, we lack insights in New York City as to which Hispanic subgroups experienced the worst risks or mortality caused by the COVID-19 pandemic.

On-going qualitative research in New York City by the lead author [4] suggests that Hispanic families in households with mixed immigrant status experienced the worst forms of economic and health vulnerabilities due to COVID-19. Accordingly, living arrangements, family structures, education, and years of residence help explain the higher rates of infection, hospitalization, and deaths in these households. These preliminary insights also suggest that Hispanic women living with a spouse or male partner experienced higher risks of COVID-19 infections and hospitalizations than those who headed home on their own. In addition, emerging COVID-19 literature has argued that, nationally, while women experienced labor market shocks, losing jobs at much higher rates than men, the latter paid a higher price with their lives [41]. An established epidemiological scholarship also documents that while women tend to experience higher comorbidities and a lower quality of health, especially as they age, men tend to develop more chronic illnesses and die sooner. These findings contribute to what has been conceptualized as a gender health paradox [42]. The results have led these and other scholars to advocate for inter-disciplinary studies bridging public health and social sciences scholarships to distinguish the main factors affecting men’s and women’s health choices, resources, and outcomes. Beyond structural drivers of discrimination, implicit bias by health care providers based on gender, race, ethnicity, and other characteristics is now associated with the exclusion or misdiagnosis of women and those of other minority groups’ health conditions [61].

Given the higher than usual COVID-19 death rates among different populations in specific spatial locations, researchers have also emphasized that we pay attention to geography in explaining the higher COVID-19 risks and death rates among the US Hispanic population [3]. This is particularly the case in specific regions where the Hispanic population has traditionally concentrated [5,6,60]. The higher-than-average death rates among Hispanic groups are surprising, as close to four decades of studies have consistently documented the health advantages of foreign-born Hispanics when compared to whites or their native-born counterparts. The phenomenon has been conceptualized as the Hispanic health paradox (HHP) [2,7,8,9,10,22]. However, researchers now argue that the COVID-19 pandemic has eroded the health advantage of Hispanics given the higher death rates experienced by the group since the pandemic.

Beyond geography, scholars have also examined the role of structural racism [11], one especially exacerbated by the nesting effects of intersectional inequalities the Hispanic population confronts in the US, such as concentration in low-wage service occupations with higher risk of COVID-19 exposures [6]; limited access to health care services [9] such as vaccinations [6] or preventive care; and the weathering effect of living in highly concentrated, poor, inner-city neighborhood areas [11].

Studies have also shown that the positive correlation observed between death rates and the concentration of Hispanics seems to vanish at the county level when socio-economic characteristics linked to structural racism are controlled [12]. Ample research has documented the role of geography on life expectancy [13,14], the rate of COVID-19 spread [15], and the relation of air pollution concentration to deaths because of COVID-19 [16]. Most significantly, these emerging insights reveal that COVID-19 deaths are linked to spatial location at the county level [17]. However, other studies [17] based on national data at the county level find no spatial correlation between COVID-19 deaths and the Hispanic population. Yet, our analysis of panel data at the level of the unit of metropolitan areas [PUMA] for New York City finds a positive spatial correlation between the Hispanic population’s COVID-19 death rate during the peak year of the pandemic. This result is consistent with previous studies, also based in New York City, documenting the correlation between neighborhood characteristics and spatial distribution in determining the health needs of specific groups [18].

Informed by the above scholarship, this paper further examines how structural racism (measured by spatial segregation) and intersectional forms of gendered inequality (measured by family structure, employment, and citizenship status) combine to explain the disproportionally higher rates of COVID-19 deaths among Hispanic groups in New York City during the first and most devastating year of the pandemic. Beyond the analysis of spatial correlations, we document differences in gendered vulnerabilities associated with the risk of Hispanic men and women dying from COVID-19. An important caveat in our study is that once we control for neighborhood (PUMA) characteristics, the positive correlation between the death rate and the Hispanic population vanishes for women but not for men. This result is new and invites further research by gender, spatial stratification, and public health scholars, especially among proponents of the Hispanic health paradox. Our analysis of different subsamples reveals that Hispanic men with a shorter time of residence in the U.S., citizenship status, and health insurance have better health outcomes than those with a longer residence. We interpret the importance of the time spent in the US as proof that the weathering process matters [11] in decreasing the health advantage of Hispanic groups. We also interpret the importance of health insurance and citizenship status as class indicators increasing access to preventive health services, such as vaccination [6].

We perform further analysis focused on women to understand what characteristics of the PUMA data help eliminate the positive correlation between COVID-19 deaths and the Hispanic population share. A separate, ongoing qualitative study by the lead author [4] in New York City suggests that Hispanic immigrant women who import at least one year of higher education and who act as sole heads of household experience lower risks of COVID-19 infection and deaths. Our quantitative analysis reveals that for different subsamples of women, only the inequality variables of the Townsend index explain in all the subsamples the positive correlation between deaths and the Hispanic population. These variables include the share of the Hispanic population that lives in overcrowded households, uses public transportation, is unemployed, has children, and lives in poverty.

This paper is organized as follows: the first part discusses our theoretical considerations of the Hispanic health paradox, structural racism, gender inequality and its relation to health outcomes, and the applicability of this scholarship for framing our findings on the role of spatial segregation and social inequality on the disproportionate death rates observed for the Hispanic population in New York City; the second part of the paper introduces our empirical models used to calculate death differentials for the Hispanic population; the third part presents the data and results of findings; the fourth part presents our discussion and conclusion as well as recommendations for future research.

## 2. Theoretical Considerations

### 2.1. The Hispanic Health Paradox

Ample research has documented the health advantage of foreign-born Latinos when compared to the non-Hispanic, native-born population [7,8,9,10]. The Hispanic health paradox (HHP), for example, explains the observed higher-than-average life expectancy among the Hispanic population despite their lower-than-average socioeconomic characteristics. It is termed a “paradox” because, despite their lower socioeconomic position, Hispanics, when compared to whites, have better health outcomes on some measures such as mortality, cardiovascular disease, and premature births [19,20] than native-born white groups. The “paradox” is especially pronounced for first-generation immigrants and those from Mexico and parts of Central and South America [21,22,23].

However, even prior to the coronavirus pandemic, scholars had already noted several health trends challenging the HHP, especially among Mexican Americans. The U.S.’s Mexican American population exhibits higher obesity rates, higher Type 2 diabetes rates, higher absolute increases in the prevalence of diabetes, and much higher mortality rates due to diabetes compared to the white population [24,25,26]. A recent study [59] in El Paso, Texas, reveals that a high share, or 49.3%, of Hispanic respondents had been previously diagnosed with a high-risk co-morbidity and that this history increased their likelihood of hospitalization or dying from COVID-19-related illnesses. Additional forms of inequalities, such as poverty, homelessness, and the lack of citizenship status and medical insurance, increased these risks. Other research has evidenced that Hispanics in general also exhibit higher rates of disability, lower rates of health insurance, and lower access to quality health care [27,28]. Many of these studies also show that the health advantage of Hispanic immigrants declines significantly the longer they reside in the US, despite increases in socioeconomic status, with middle-class, native-born Hispanics reporting higher risks of non-communicable disorders, such as diabetes and stroke, even when compared to their lower-SES co-ethnics and the white population [14,29]. Beyond our findings’ contribution to the revisiting of the Hispanic health paradox, we contribute a new analysis focused on spatial segregation], as a measure of structural racism, to understand the correlation between spatial concentration and health outcomes and explain the determinants of the higher death rates experienced by New York City Hispanics.

### 2.2. Structural Racism and Health Outcomes

A legacy of structural racism [30] (p. 1457) put in place with the US institutions of slavery and Jim Crow Laws, leading to the institutionalization of racist practices and ideologies, has benefited the life chances of majority white groups at the expense of African Americans, Hispanics, and other people of color in this nation [31]. There is also evidence that, beyond structural inequality resulting from overt racial discrimination, implicit racism, which is harder to measure, combines to negatively affect the quality of health services non-white individuals receive [32,34,61]. Implicit racism has been defined as ethnic or racial bias impacting the behavior of health care providers in the delivery of care. It is not “consciously acknowledged and operates through subtle ways” [33] (p. 44). Although focused mainly on measuring structural racism through spatial segregation, our findings also have implications for the added effect of implicit racism or discrimination in increasing the vulnerabilities of COVID-19 contagion, hospitalization, and deaths for Hispanic groups in specific neighborhood areas of New York City, as our findings will reveal.

### 2.3. Structural Inequality, Sexism and the Health Outcomes of Women and Men

An established literature has documented that gender discrimination, or sexism, is a significant determinant of lower health outcomes and higher rates of comorbidities for women than for men [49,50] However, most studies focused on explaining discrimination along gender lines are focused on individual effects or aggregates at the group level [50,51]. More recent scholarship has argued that the health discrimination women experience is multilayered and reinforced by institutions such as the labor market, the political system, and even the health care system [49,52]. For example, lower wages and limited access to insurance and gainful employment have proven to negatively impact women’s access to quality and timely care delivery, increasing their risk of discrimination within the health care system. Emerging COVID-19-focused research now also documents that intersectional forms of gendered and race-based inequalities combine to impact health, which is defined as “the totality of ways in which societies foster racial discrimination through mutually reinforcing systems of housing, education, employment, earnings, benefits, credit, media, health care, and criminal justice [44,46]. These patterns and practices, in turn, reinforce discriminatory beliefs, values, and the distribution of outcomes of hospitalized COVID-19 patients, with white minority groups experiencing the higher adverse risks [53,58].

A recent U.S. study focused on structural inequality and gendered disparities in health care access and health outcomes [47,49] traces the impact of state sexism on the lower health outcomes of women. Drawing on the national consumer survey of health care access, the scholars find that in states with more widespread access to medical insurance, such as Medicaid, including resources from the Safety Net, women enjoy better health outcomes. Most importantly, structural sexism is a strong predictor of higher rates of morbidity and mortality for women, as in states with higher levels of economic inequality, women and children experience higher rates of mortality [48,50,51]. The opposite is also observed. In states where women experience lower institutional sexism, women have fewer barriers to care and access to more comprehensive health care than men. Drawing on this literature, our paper explores how structural and social intersectional forms of inequalities combine to affect the higher COVID-19 death rates among subgroups of Hispanic men and women in New York City.

### 2.4. Structural Racism and the Fates of Hispanic Groups during the COVID-19 Pandemic

The higher death rate due to COVID-19 reported among the Hispanic population [3,5,6] has led researchers [2] to conclude that the health advantages the group had enjoyed have been reduced by the ravages of the COVID-19 pandemic and ensuing decimation of the population. However, scholars disagree about the plausible causes. One scholar [11] attributes the higher death rate to structural racism, specifically to three characteristics observed among the Hispanic population: (a) a higher risk of COVID-19 exposure; (b) lower quality of health care and limited access to health insurance; and (c) the negative and cumulative impact of a weathering process with more years of residence and acculturation in the U.S. [8]. Another scholar [6] has argued that the occupations of the Hispanic population and the lower vaccination rates among the group also contribute to higher-than-average death rates. Yet, the qualitative insights of a longitudinal qualitative study by the lead author and her colleague [4] reveal that Hispanic immigrants in New York City have had the highest rate of COVID-19 vaccinations among other groups. Some other authors have shown that the positive correlation observed between COVID-19 death rates and the share of the Hispanic population at the county level vanishes when the socioeconomic characteristics of the county, which are linked to structural racism, are controlled [12]. We further explore this analysis in the paper.

### 2.5. Spatial Concentration and Death Rates

Studies have documented the existence of spatial concentration patterns in the spread of COVID-19 cases and deaths. One study [15] shows that over time, key nodes of regional concentration and expansion of the disease formed mostly within five metropolitan areas in the continental U.S.: New York, Chicago, Los Angeles, Miami, and Houston. These scholars argue that these five regional nodes formed due to different economic, social, and environmental factors. However, other scholars [16] have shown that deaths concentrated within metropolitan areas are due to the correlation between high pollution and the severity of the COVID-19 disease. Interestingly, among the top metropolitan areas identified as having the highest concentration of Hispanic populations, New York continues to rank among the top cities where Hispanics experience the highest indexes of spatial segregation and inequality [55].

Bronx County, for example, has the largest share of Latinos in New York State [17,55]. In explaining the impact of structural inequality on the higher death rates of U.S. minorities, scholars have analyzed spatial concentration at the Bronx County level. They find a significant correlation between spatial segregation and deaths for non-Hispanic black groups but not a significant correlation between spatial segregation and death rates for Hispanics. However, our more focused analysis of spatial concentration at the neighborhood level in New York City shows that spatial segregation increased the risk of death for Hispanics in the Bronx. We also explore how different forms of gender and class inequality, measured by education, employment, and citizenship, combine to further increase the vulnerability of COVID-19 infections, hospitalizations, and deaths among different Hispanic subgroups.

## 3. Methodology

As a complete data set that disaggregates deaths by PUMA (Public Use Microdata Areas (PUMAs) are non-overlapping, statistical geographic areas that partition each state or equivalent entity into geographic areas containing no fewer than 100,000 people each) by race and ethnicity is not available, researchers have looked at the correlation between specific population shares and the total number of deaths [12,35,36]. In this paper, we follow the approach of McLaren [12] to indirectly estimate the differential mortality ratio between Hispanics and the rest of the population. The estimation of this mortality differential is the first objective of this paper. The method consists of estimating the partial correlation between total deaths and the Hispanic population, as shown in Equation (1):
(1)$$D_{i}=\beta_{0}+\beta_{1}hispanic_{i}+u_{i}$$
where $D_{i}$ represents deaths at PUMA *i*; $hispanic_{i}$ represents the share of Hispanic population in PUMA *i*; $u_{i}$ represents the usual error term; and $\beta_{1}$ represents the correlation between deaths at PUMA level and the Hispanic population share. Following [12], define the weighted average deaths in PUMA *i* by Equation (2):
(2)$$D_{i}=hispanic_{i}d_{h,i}+\left(1-hispanic_{i}\right)d_{-h,i}$$
where $d_{h,i}$ represents deaths among Hispanics in PUMA *i*, and $d_{-h,i}$ represents deaths among the rest of the population in PUMA *i*.

Differentiating Equations (1) and (2) with respect to $hispanic_{i}$
(3)$$\frac{dD_{i}}{dhispanic_{i}}=\beta_{1}$$
(4)$$\frac{dD_{i}}{dhispanic_{i}}=d_{hispanic,i}-d_{-hispanic,i}$$

Which implies that:
(5)$$\beta_{1}=d_{h,i}-d_{-h,i}=\Delta_{h}^{*}$$

$\Delta_{h}^{*}$ is the death mortality differential for the Hispanic population. If $\beta_{1}>0$, the Hispanic population has a higher mortality ratio than the rest of the population. Studies usually introduce variables that can measure structural racism. If the mortality differential does not disappear, then the residual differential is not explained by structural racism [11,34]. If $\beta_{1}<0$, we would argue that the Hispanic population has a lower death prevalence, which is explained by the Hispanic health paradox [8,9,10].

Equation (1) suffers from biases generated by a model that is not well specified because of the nature of the data and the existence of more covariates. Consequently, we include more control variables that could explain the mortality differentials, taking into consideration the nature of the data. Given the panel data structure of our data for deaths and the cross-section nature of the data on the PUMA characteristics, we estimate a random effect estimation on the change in death rate by PUMA:
(6)$$\Delta D_{it}=\beta_{0}+\beta_{1}hispanic_{i}+\sum_{k=1}^{K}\gamma_{i}X_{ik}+u_{it}$$
where $X_{ik}$ represents control variable k. As control variables, we include the share of the population with some college or more education, the share of the population with high school completed, the share of the population working in 13 specified occupation categories, the median household income, the share of the insured population, the share of the population that drives to work, the share of the unemployed population, the share of the population that rents, the share of overcrowded households, and the fraction of households with child poverty. These variables are linked to structural racism [12] and the Townsend Index [36,43], a deprivation index that has been found to be correlated with geographic areas with high health problems [35]. This specification has the advantage of eliminating any potential time trend existing in the death rate.

Equation (7) shows a second specification where we exploit the spatial distribution of the data to consider the potential spatial distribution of deaths [15,16]. Specifically,
(7)$$\Delta D_{it}=\beta_{0}+X_{i}\beta +\bar{X_{i}}\gamma +\lambda W\Delta D_{it}+\rho Mu_{i}+\epsilon_{it}$$
where $X_{i}$ includes all control variables and the Hispanic population, $\bar{X_{i}}$ includes the spatial lags for control variables, W and M are spatial weighting matrices, and λ and ρ are scalar parameters. In this specification, each variable is said to have a direct and an indirect effect. Direct effects refer to impacts that are not linked to spatial correlation, while indirect effects are linked to the different sources of spatial correlation that are present in Equation (7).

Exploiting the spatial panel nature of the data has certain advantages over an OLS estimation. First, by considering the spatial correlation, we obtain a better estimation than the one offered by OLS. Second, the spatial specification allows us to obtain direct and indirect effects, which allows the determination of the importance of spatial correlations in determining the effects of each variable.

The second analytical objective of this paper is to obtain gendered estimations for Equations (1), (6), and (7). The data on deaths is aggregated; consequently, we cannot separate it by gender. The data at the PUMA level, however, comes from the American community survey (ACS) and allows for the calculation of characteristics by gender by aggregating the PUMA characteristics only for the population of adult men and women, respectively.

A third objective of the paper is to further analyze the origins of the results. We will do so by generating different subsamples at the PUMA level, which will allow us to do a comparison of the estimations of Equations (1), (6), and (7) for the different partitions of the data. The analysis is explained fully in Section 4.4.

## 4. Results

### 4.1. Data Sources

Zip code data on mortality comes from New York City health [NYCH], which provides COVID-19 cases and deaths by zip code [37]. The data was collected for the months of November 2020 through June 2021. Data was transformed to PUMA level using a cross walk available from Baruch College’s Newman Library [38].

As Figure 1 shows, the average death rate by PUMA for the period of November 2020 to June 2021 shows a positive spatial correlation with the Hispanic population in New York City. The dark blue areas indicate a higher concentration of the Hispanic population, with the highest share concentrated in the Bronx and Queens, where death rates for the group have been highest. Our analysis shows that this correlation is statistically significant at the 1 percent level with a chi squared of 10.81 and with 1 degree of freedom.

The significance of the spatial correlation contrasts with results reported by Saffary et al. [17] at the county level for the entire US, since they report that the spatial correlation between county COVID-19 death rate and Hispanic population is not significant.

Table 1 shows the average death rate per PUMA for the period beginning in November 2020 and ending in June 2021. It clearly shows a positive increase in the death rate during the specified time period. 

The data for the Hispanic population was obtained from the 2019 American community survey (ACS) 5-year sample obtained from Public Use Microdata Areas (PUMAS) [39]. The data selected were men and women 18 years of age and older. Table 2 shows the average values for the different variables obtained from this source. In particular, the median income in 2019 was USD 29,000, the Hispanic population was 20% of the population, 56% of the population have high school or more education, 22% of the population have college or more education, 92% of the population have health insurance, 26% of the working-age population drive to work, 3.7% of the population are unemployed, 12% of the population have an occupation in management related activities, 3.9% have an occupation in computer-related activities, 11% of the population have activities related to education, 3.9% of the population have activities related to health services, 14.2% of the population work in other service occupations, 7% of the population work in sales, 8.7% work in offices, 0.1% in farming, 2.8% in construction, 1.4% in installation, 2.3% in production, 4.6% work in transportation, and 0.1% in the military. The table also shows that 45% of the population rents, 75% live in overcrowded conditions or have more than one individual per room, and 21% of households have children living below or at the poverty line.

### 4.2. Random Panel Estimation for Monthly Death Rates

Table 3 presents the estimations for Equations (2), (6), and (7). In the three equations, we use the change in the logarithm of the death rate as the dependent variable. Column (1) in Table 3 shows a negative correlation between the change in the COVID-19 death rate and the Hispanic population. This result has the opposite sign from that explained in the previous section. This shows that the positive correlation between deaths and the Hispanic population exists at certain levels, but at the first difference, the correlation becomes negative, which is consistent with the predictions advanced by the Hispanic health paradox for the group. This result, however, is not robust to the inclusion of the characteristics of the PUMAs, or to the spatially correlated terms. These results imply that when we control for structural racism (based on the characteristics of the PUMAs), no correlation is found between deaths and the Hispanic population.

Table 3 also shows in Column (2) that in the random effect estimation, a negative correlation is found between the share of individuals that drive to work and deaths, which is expected given that people driving to work would maintain social distance. Table 3 also shows a negative correlation between the share of individuals that work in health services and death rates, which probably shows that individuals in the health industry were more aware of taking measures to avoid infections. Table 3 shows a negative correlation between the share of individuals working in construction and deaths, probably because individuals working in construction stop working and therefore avoid risks. Table 3 also shows a negative relationship between people who rent and deaths, probably showing that houses under rent had better conditions that avoided infection risks, such as lower levels of overcrowding. Finally, PUMAs with a larger share of children living in poverty show a negative correlation with the death rate, perhaps because in households with younger individuals, there were fewer deaths or because individuals in these households were forced to remain at home to take care of small or school-age children when schools closed or transitioned to online education, reducing the risk of COVID-19 contagion.

Columns (3), (4), and (5) in Table 3 show the results for the spatial random effect model. They show the direct, indirect, and total effects estimated for each variable. This model shows different results as compared to the random effect model. Now, the share of individuals with high school or more years of education has a direct negative effect on deaths. No indirect effect or total effect is found. This result probably shows that individuals with high school or higher education were less exposed to the disease, had access to resources, including vaccines, or had the education to obtain information about timely prevention.

Table 3 shows that having a college education or more has a positive direct effect on deaths, while no indirect or total effect is found to be significant. This result shows that, on average, Hispanics with a college education or more were more exposed to the disease, as these individuals’ jobs probably required more social or direct interactions with the public or people they serve. This result is correlated with the qualitative findings from ongoing research conducted by the lead author and her colleague in New York City [4].

Table 3 shows that the share of the population with health insurance is negatively correlated with deaths in both direct and total effects. No indirect effect is found. This result is important and in line with Garcia et al. [11], who find that structural racism explains the disproportionate impact of the COVID-19 pandemic on black and Latinx people since the lower availability of health insurance among the Hispanic population is mentioned as a risk factor.

Table 3 also shows that the share of the population that drives to work has direct, indirect, and total negative effects, illustrating the importance of social distancing for avoiding infections and deaths. Table 3 also captures a negative direct effect for the share of the population that works in construction, but no indirect or total effect is found. The result indicates that individuals working in this sector probably were laid off or lost their jobs as the industry closed and consequently avoided risks of infections and deaths.

Table 3 shows that the share of individuals who rent has a negative direct, indirect, and total effect. This probably shows that the housing conditions for individuals in rented housing were more protective of infections and deaths. However, individuals living in overcrowded households showed a positive direct effect but not an indirect or total effect associated with deaths. This shows that individuals in overcrowded households had more difficulty avoiding infections or keeping social distance.

### 4.3. Estimations for Men and Women

Table 4 presents estimations for men and women. We first look at the positive correlation that exists between the level of COVID-19 deaths and Hispanic men, which indicates that they have 0.78 more deaths than the non-Hispanic male population. In the case of women, the excess death rate is 0.72. Once we introduce the spatial correlation, we can see that the positive direct effect of being Hispanic and male is 1.13, while the indirect effect is −1.08. The total effect is not very significant. These results imply evidence of both implicit and structural discrimination, while the effect of the spatial correlation is factually positive. This result is consistent with other findings showing that spatial concentration in New York City somehow mediated the ability of some Hispanic individuals to combat the pandemic [40]. For women, the direct effect of spatial concentration is 1.15, almost identical to the effect observed for men. The indirect effect of spatial concentration is −1.20, which is larger than the effect observed for men. The total effect of spatial concentration we find to be negative and insignificant in explaining COVID-19’s higher death rates. In fact, our findings suggest that women benefit more from living in segregated or concentrated neighborhood areas than men, and this deserves further research by spatial stratification, poverty, and public health scholars.

Table 4 also shows that once we introduce the PUMA characteristics, for men, the direct effect of spatial concentration is 0.26, while the indirect effect and the total effect are not significant. This would imply that after controlling for spatial concentration or structural racism, Hispanic men still observe an excess death of 0.26, which evidences that the higher risk of dying for Hispanic males is also explained by implicit bias. In the case of women, after controlling for the characteristics of the PUMA, the direct, indirect, and total effects remain all non-significant in explaining COVID-19’s higher death rates. This is consistent with an earlier study [12] that found that after controlling for characteristics, there was no evidence of implicit bias for Hispanics. However, our result finds that such a result holds only for Hispanic women but not for men.

### 4.4. Using Different Hispanic Subsamples: Time in the U.S., Citizenship, Employment Status, Health Insurance, Never Married, and Heads of Household

In this section, we show the results of generating further analysis among subsamples of Hispanic groups to identify potential explanations for the different levels of intersectional vulnerabilities that combined to differently increase the death rates of Hispanic men and women in New York City. The analysis uses years of residence as a proxy for nativity or for the correlation between the weathering effect or reduction of protective health factors associated with nativity or with acculturation to the US’s culture [8].

Time in the U.S. and Citizenship: Table 5 looks at subsamples generated for individuals who arrived in the U.S. since the year 2000 and for individuals who are also U.S. citizens. The first variable measures if the time spent in the U.S. is linked to mortality results, given the importance of the weathering process [11]. The weathering hypothesis implies that health outcomes should deteriorate the longer Hispanics reside in the U.S. In the case of citizenship, the argument is that it should be positively related to better health outcomes. Table 5 shows that for the two subsamples, the positive correlation between deaths and the Hispanic population is observed at the same levels for men and women. When the spatial correlation is considered, the direct effect is positive for both men and women, and the indirect effect is negative. The total effect is non-significant. If we control for the characteristics of the PUMA, no significant correlation between deaths and the Hispanic population is observed for either men or women. This would imply that the characteristics of the PUMA, and consequently of spatial concentration or structural racism, explain the excess deaths of Hispanic men and women. Considering that for the entire sample of men, we did find a positive correlation between deaths and the Hispanic population, this result would imply that individuals that have less time in the U.S. and that are citizens have better health outcomes, which consequently shows that the weathering hypothesis applies [11] and that citizenship matters. This second aspect may indicate that a vulnerable immigrant status or lack of legal documentation also negatively impacts access to preventive care for individuals living in New York City, perhaps due to a fear of apprehension in the search for medical care.

Employment and Health Insurance: Table 6 presents results for subsamples of individuals that are employed and also those with health insurance. In the case of employed individuals, the argument is that their higher exposure to the virus may deteriorate their health status [6,11]. In the case of individuals with health insurance, access to health services may explain better health outcomes [11]. Table 6 shows that there is a positive correlation between deaths and the Hispanic population for both men and women included in these subsamples. Controlling for spatial correlation renders a positive direct effect, while a negative indirect effect is found. If we control for the PUMA characteristics, in the case of the subsample individuals that are male and employed, we observe a positive correlation between deaths and the Hispanic population. The direct, indirect, and total effects are found to be significant and positive. This would imply that employment alone cannot explain the positive correlation between deaths and the Hispanic population.

In the case of individuals with health insurance, once we control for the PUMA characteristics, the correlation between deaths and the Hispanic population vanishes. This would imply that individuals with access to health insurance have better health outcomes. In the case of Hispanic women, once we control for PUMA characteristics, the positive correlation vanishes. These results corroborate emerging labor market research showing that the ravages of the pandemic led men to lose their lives, while women mostly lost their jobs, as many of these jobs concentrated in the ‘care’ or low-wage service industry, with higher levels of interaction with the public, leading to higher unemployment rates among women than men [41], and consequently, lower risks of COVID-19 infections and deaths.

Family Structure and Gendered Effects: Table 7 presents results for subsamples of individuals that have never married and that are single, as well as for individuals that are heads of households. Single individuals may have a higher risk of exposure, assuming they face more barriers to keeping social distance. Heads of households may have a lower risk of exposure, assuming they delegate who gets to carry out errands. Table 7 illustrates that for the two gender subsamples, there is a positive correlation between the level of deaths and the Hispanic population, both for men and women. If we control for spatial correlation, the direct effect is positive, while the indirect effect is negative. If we control for PUMA characteristics for the subsample of never-married men, we find that the direct and indirect effects are positive, as is the total effect. This would imply that marital status cannot explain the positive correlation observed between deaths and the Hispanic population. In the case of the sample of male heads of households, the direct effect is found to be statistically significant. This would imply that household headship alone would not explain the positive correlation observed between COVID-19 deaths and the Hispanic population. In the case of women, once we control for PUMA characteristics, the positive correlation vanishes.

Age and Weathering Effects: Table 8 presents estimations based on a comparison by age for two different Hispanic subsamples. The first subsample comprises younger individuals, between 18 and 35 years old, while the second includes older individuals, between 36 and 53 years old. According to the weathering process hypothesis [11], younger individuals should experience better health outcomes than individuals with more time in the US, which in principle should be observed among medium-aged individuals. On the other hand, if younger individuals are more likely to be undocumented, they may concentrate in jobs with higher risks of exposure [6,11] or have lower access to health insurance [11]. Table 8 shows that a positive correlation is found at levels between the Hispanic population and deaths, both for men and women. Once we control for spatial correlation, the direct effect is found to be significant, the indirect effect is negative, and the total effect is non-significant. This would imply that spatial concentration actually helps reduce the impact of the pandemic [40]. If we control for PUMA characteristics and differentiate the data, for young men, a positive total effect is found, while for young women, the correlation vanishes. In the case of middle-aged men, the correlation vanished, while for medium-aged women, the correlation became negative, confirming the prediction of the weathering effects, which are foundational for proponents of the Hispanic Health Paradox.

### 4.5. Using Less Control Variables for Women

Table 9 presents estimations for the different subsamples of women, using less control variables than those used in Table 5, Table 6 and Table 7.

Additionally, for the full sample of women, Table 9 shows that controlling for occupations is not enough to explain the positive correlation between deaths and the Hispanic population. On the contrary, median income, education, and the variables of the Townsend index are enough to explain by themselves the positive correlation between COVID-19 deaths and the Hispanic population. Similar findings are observed for the subsample of women with health insurance and the subsample of citizens. For the subsample of recently arrived women, neither median income nor education can explain the positive correlation between deaths and the Hispanic population. Only occupations and the variables of the Townsend index can explain this positive correlation. For the subsample of heads of household that are women and employed, median income cannot explain the positive correlation between deaths and the Hispanic population. However, occupations, education, and the variables of the Townsend index can explain the effect of structural racism.

For the subsample of women who are never married and single, the occupations, the median income, the education, and the variables of the Townsend index can explain structural racism. These results imply that these women have better health outcomes and that structural racism can be explained by almost any of the PUMA characteristics.

## 5. Discussion

This paper studies the relationship between death rates at the PUMA level and the Hispanic population of New York City. Our census-based analysis confirms results by other scholars [3,60] documenting a positive correlation between COVID-19 death rates and the share of the New York City population that is Hispanic. Our analysis further shows that the death rate is spatially correlated with the Hispanic population, which offers a new and more varied understanding than is provided by earlier studies [12,17]. In addition, when we control for PUMA neighborhood characteristics on death rates, the positive correlation between Hispanic population and death rates vanishes for women but not for men. The first result confirms findings by McLaren [12], while the second gendered result for women is new. This implies that in neighborhoods with a high concentration of Hispanic population, women may experience higher protective factors that decrease the risks of dying from COVID-19, while we did not observe this for men. Some of these factors may be mediated by women’s access to more diverse networks among people with longer tenure in the US or within local or mainstream institutions, such as schools and migrant-servicing agencies. This in turn increases access to health-preventive information or even access to the safety net.

Different subsamples of Hispanic individuals from different age groups confirm the results of our analysis for women. In fact, for medium-aged women, we find evidence of the Hispanic Health Paradox, since a negative correlation between deaths and the Hispanic population is found among individuals with longer tenure in the U.S. In the case of men, individuals with less time residing in the U.S., access to medical insurance, and U.S. citizenship tend to have better health outcomes than those with more years of residence in the U.S. These results substantiate earlier studies [4,11], except for our finding about the higher risk of death for young Hispanic men, which is a new finding in the literature. This may be due to younger men being more likely to live in neighborhood areas with higher population concentration, which exert greater traditional expectations on them as providers; or these men may be concentrated in jobs with higher exposure or less protective factors, such as access to health insurance. These risks translate to higher COVID-19 vulnerabilities for younger or immigrant males in more recent cohorts.

In the case of women, different specifications of the model are analyzed for the different subsamples. We find that in all subsamples, the variables of the Townsend index explain the positive correlation between deaths and the Hispanic population, which confirms results found by McLaren [12] and other scholars in other geographic areas where the Hispanic population concentrates [60]. In the case of the occupations, we find that they can explain the positive correlation between subsamples of women that are recent immigrants, heads of households, employed, and never married/single, which confirm results found by García et al. [11] and Riley et al. [6]. The median income can explain the positive correlation between deaths and the Hispanic population for those with citizenship status, for women with health insurance, and for those never married/single, confirming results by EDQ [40]. The variables for education can explain the positive correlation between deaths and the Hispanic population in the case of heads of household, women who are employed, have health insurance, are citizens, and those never married/single, confirming results by EDQ [40]. These results also imply that women who have never married and are single are over-represented among the subsample of Hispanic individuals with better health outcomes. These results confirm those by Fuentes and Kucheva [4].

## 6. Conclusions

The first objective of this paper was to estimate the correlation between deaths caused by COVID-19 and the Hispanic population. If the estimation is performed at neighborhood levels, our results show a positive correlation between deaths and the Hispanic population, revealing a larger death rate among the Hispanic population when compared with the rest of the population of New York City. If the estimation is performed among subgroups, we obtain a negative correlation between deaths and the Hispanic population, substantiating the predictions of the Hispanic Health Paradox. If we control for spatial correlation effects and variables related to structural racism, the correlation between deaths and the Hispanic population vanishes.

The second objective of the paper is to estimate the correlation between deaths and the Hispanic population separately for men and women, controlling for the gendered characteristics of the PUMA. The results show that the same results obtained for the entire sample are obtained for women, while for men, we find a positive direct effect that is offset by the indirect effects of spatial correlation, rendering a non-significant total effect. For men, four elements are found to negatively affect their health in the U.S.: (1) the time spent in the U.S.; (2) their employment status; (3) their civil status or never married/single, and (4) their young age. Three elements are found to positively affect health outcomes: (1) US citizenship, (2) access to health insurance, and (3) being of medium age. In the case of women, only the characteristic being of medium age is found to positively affect the health status of individuals. In all other cases, the control variables linked to structural racism explain the positive correlation observed at different levels.

These findings reveal that women, despite suffering infections from COVID-19, had lower death rates, as has been found for all women nationally [50]. We argue that these results highlight the need for additional studies to better understand why Hispanic women seem to derive a more protective effect from living in areas with a higher concentration of the population. Women appear to be more resilient to COVID-19, particularly when the results are linked to the occupations they perform, their access to better health services, or other biomedical reasons for these results. The results also highlight the need for improving the health services for the immigrant young men population of New York City, since they seem to perform riskier and essential jobs for the city and to experience higher limitations or vulnerability when accessing health insurance, and as research shows, they died at higher rates during the pandemic. Our results also suggest the need for further research on the needs of older immigrant adults, such as women, since we find evidence that the weathering hypothesis continues to matter or to shorten their presumed immigrant health advantages. Our results also emphasize that it is imperative that federal, state, and local governments collect and release comprehensive data on the number of confirmed COVID-19 cases and deaths by race/ethnicity and immigrant status, as well as age, to better gauge the impact of the outbreak across non-white Hispanic groups in mixed-status immigrant households. Finally, we claim that our results also highlight the need for a discussion among academics and policymakers of the different incremental steps to be taken to reduce the disproportionate burden of other potential future pandemics among minorities, given the decimation that the pandemic has caused among older black and vulnerable Hispanic populations in the U.S. Our results mainly emphasize the need for transformative actions that address structural racism to achieve greater health equity among historically racialized U.S. minorities. Finally, as Hispanics and Latinos make up a disproportional share of the young U.S. population, projected by 2030 to compose 22 percent of the U.S.’s labor force [1], the size and composition of the population and its contribution to the U.S. economy make it more imperative to design policy solutions to reduce structural racism and other forms of social inequalities that threaten the recovery of the group from the ravages of the pandemic and its future members contributions to our society.

## Figures and Tables

**Figure 1 ijerph-20-05838-f001:**
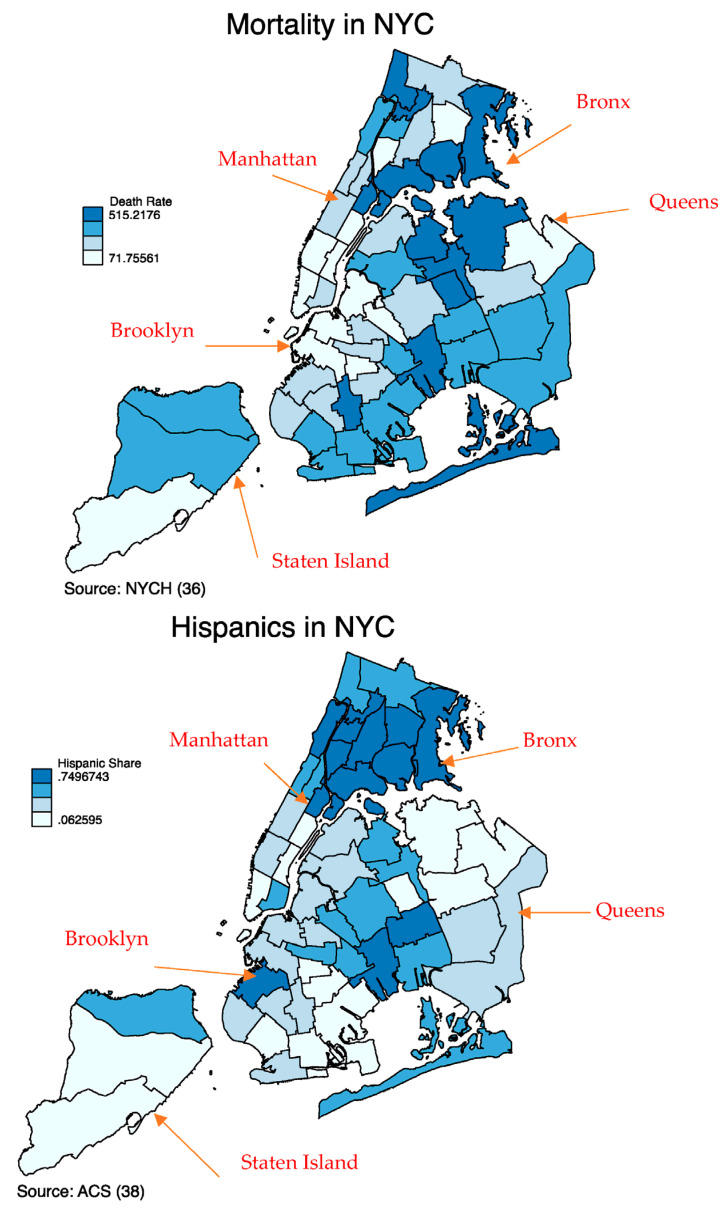
Source: Own calculations with data from NYCH [37] and ACS [39].

**Table 1 ijerph-20-05838-t001:** COVID-19 Death rate in 55 NYC PUMAs.

Period	Mean	S.E.
November 2020	186	78
December 2020	186	78
January 2021	195	79
February 2021	216	86
March 2021	236	92
April 2021	253	97
May 2021	263	100
June 2021	267	102

Source: own estimations based on [37].

**Table 2 ijerph-20-05838-t002:** Mean values for selected variables.

	Mean	S.E.
Median income	29,746	14,486
Shares		
Hispanic	20.4%	15.1%
High School completed	56.1%	12.8%
College or more	22.4%	7.3%
Health Insurance	92.5%	2.9%
Drive to work	26.6%	13.3%
Unemployed	3.7%	0.9%
Management	11.8%	5.7%
Computers	3.9%	1.6%
Education	11.0%	3.9%
Health	3.9%	1.2%
Services	14.2%	4.2%
Sales	7.0%	1.1%
Office	8.7%	1.4%
Farming	0.1%	0.1%
Construction	2.8%	1.2%
Installation	1.4%	0.6%
Production	2.3%	1.1%
Transportation	4.6%	1.7%
Military	0.1%	0.4%
Rent	45.3%	15.0%
Overcrowded	74.8%	8.1%
Child poverty	21.0%	9.3%

Source: 5-year public sample, 2019 American Community Survey (ACS) [39].

**Table 3 ijerph-20-05838-t003:** Results for RE and spatial RE models. NYC PUMAs, November 2020–June 2021.

Model	RE Model	RE Model	Spatial RE Model		
			Direct Effect	Indirect Effect	Total Effect
Hispanic	−0.03 *	0.06	0.24	2.14	2.39
	[0.015]	[0.04]	[0.16]	[2.07]	[2.22]
Median income		−4.4 × 10^−7^	2.4 × 10^−7^	−3.3 × 10^−6^	−3 × 10^−6^
		[7.5 × 10^−7^]	[3.8 × 10^−6^]	[3.8 × 10^−5^]	[4.2 × 10^−5^]
High School completed		−0.15	−0.75 **	−5.68	−6.42
		[0.13]	[0.37]	[3.82]	[4.16]
College or more		0.13	1.04 *	9.80	10.84
		[0.19]	[0.58]	[6.46]	[7.01]
Health Insurance		−0.25	−1.09 ***	−5.20	−6.29 *
		[0.21]	[0.41]	[3.46]	[3.83]
Drive to work		−0.14 **	−0.45 **	−5.13 *	−5.58 *
		[0.07]	[0.20]	[3.16]	[3.34]
Unemployed		−0.09	1.27	11.52	12.79
		[0.52]	[2.56]	[19.04]	[21.49]
Management		−0.18	−0.84	−6.42	−7.26
		[0.30]	[1.12]	[10.48]	[11.54]
Computers		0.46	1.62	13.41	15.03
		[0.55]	[2.13]	[20.21]	[22.27]
Education		−0.26	−0.11	0.96	0.84
		[0.17]	[0.29]	[2.74]	[2.95]
Health		−0.79 *	0.01	22.57	22.57
		[0.47]	[1.68]	[20.64]	[22.26]
Services		−0.22	−0.13	−0.03	−0.17
		[0.20]	[0.47]	[3.99]	[4.44]
Sales		−0.42	−1.56	4.83	3.28
		[0.43]	[1.26]	[12.71]	[13.79]
Office		0.31	0.22	2.71	2.93
		[0.32]	[0.76]	[8.41]	[9.11]
Farming		0.23	16.62	201.52	218.14
		[4.41]	[15.21]	[157.51]	[171.83]
Construction		−1.09 *	−4.03 **	−23.12	−27.15
		[0.59]	[2.09]	[18.13]	[20.14]
Installation		−1.68	−4.87	−31.74	−36.62
		[1.20]	[4.32]	[41.02]	[45.11]
Production		−0.05	0.40	4.94	5.34
		[0.85]	[2.90]	[27.34]	[30.00]
Transportation		−0.42	−0.27	4.56	4.29
		[0.51]	[1.54]	[13.94]	[15.20]
Military		0.97	1.09	3.94	5.03
		[0.64]	[1.59]	[14.91]	[16.33]
Rent		−0.10 *	−0.33 **	−3.06 *	−3.40 *
		[0.06]	[0.15]	[1.82]	[1.95]
Overcrowded		0.11	0.55 *	3.14	3.69
		[0.07]	[0.32]	[3.39]	[3.70]
Child poverty		−0.18 **	−0.40	−1.34	−1.74
		[0.09]	[0.41]	[3.31]	[3.71]
Constant	0.06 ***	0.54	Na	Na	Na
	[0.004]	[0.28]			
N	385	385	385		
Pseudo R2	1%	8.40%	12.64%		

*** Significant at 1%, ** Significant at 5%, * Significant at 10%. Source: own calculations with data from [37,39].

**Table 4 ijerph-20-05838-t004:** Excess deaths for Hispanics, spatial correlation, and gendered effects.

	Direct Effect	Indirect Effect	Total Effect
No controls, no spatial correlation, levels			
Men	N/A	N/A	0.78 **
			[0.36]
Women	N/A	N/A	0.72 **
			[0.36]
With no controls, only spatial correlation, levels			
Men	1.13 ***	−1.08 *	0.06
	[0.42]	[0.62]	[0.52]
Women	1.15 ***	−1.20 **	−0.04
	[0.42]	[0.61]	[0.50]
With controls, spatial correlation, changes			
Men	0.26 **	1.44	1.70
	[0.11]	[1.29]	[1.39]
Women	−0.07	−0.13	−0.19
	[0.12]	[1.72]	[1.83]

Source: own calculations with data from [37,39]. *** Significant at 1%, ** Significant at 5%, * Significant at 10%.

**Table 5 ijerph-20-05838-t005:** Excess deaths for Hispanics, spatial correlation, and gendered effects.

	Recently Arrived Immigrants			Citizens		
	Direct Effect	Indirect Effect	Total Effect	Direct Effect	Indirect Effect	Total Effect
No controls, no spatial correlation, levels						
Men	N/A	N/A	0.61 **	N/A	N/A	0.79 **
			[0.30]			[0.39]
Women	N/A	N/A	0.54 *	N/A	N/A	0.73 **
			[0.30]			[0.38]
With no controls, only spatial correlation, levels						
Men	0.80 **	−0.65	0.16	1.21 ***	−1.24 *	−0.04
	[0.35]	[0.51]	[0.42]	[0.46]	[0.67]	[0.56]
Women	1.10 ***	−1.15 **	−0.06	1.20 ***	−1.33 **	−0.13
	[0.42]	[0.59]	[0.44]	[0.44]	[0.64]	[0.53]
With controls, spatial correlation, changes						
Men	−0.10	−0.28	−0.38	0.17	0.56	0.73
	[0.11]	[0.86]	[0.97]	[0.13]	[1.41]	[1.52]
Women	0.10	0.75	0.84	−0.03	−0.32	−0.36
	[0.08]	[0.79]	[0.86]	[0.11]	[1.32]	[1.42]

Source: own calculations with data from [37,39]. *** Significant at 1%, ** Significant at 5%, * Significant at 10%.

**Table 6 ijerph-20-05838-t006:** Excess deaths for Hispanics, spatial correlation, and gendered effects.

	Employed Individuals			Individuals with Health Insurance		
	Direct Effect	Indirect Effect	Total Effect	Direct Effect	Indirect Effect	Total Effect
No controls, no spatial correlation, levels						
Men	na	na	0.78 **	Na	na	0.78 **
			[0.34]			[0.37]
Women	na	na	0.75 **	Na	na	0.73 **
			[0.34]			[0.36]
With no controls, only spatial correlation, levels						
Men	1.03 ***	−0.81	0.22	1.46 **	−1.51 **	−0.06
	[0.40]	[0.57]	[0.47]	[0.48]	[0.69]	[0.55]
Women	1.13 ***	−1.00 *	0.13	1.49 **	−1.60 **	−0.12
	[0.44]	[0.60]	[0.47]	[0.48]	[0.67]	[0.53]
With controls, spatial correlation, changes						
Men	0.28 ***	1.78 *	2.06 **	0.22	0.94	1.16
	[0.08]	[0.92]	[0.99]	[0.14]	[1.58]	[1.72]
Women	−0.03	−0.38	−0.41	−0.06	−0.70	−0.76
	[0.12]	[1.13]	[1.25]	[0.09]	[1.31]	[1.39]

Source: own calculations with data from [37,39]. *** Significant at 1%, ** Significant at 5%, * Significant at 10%.

**Table 7 ijerph-20-05838-t007:** Excess deaths for Hispanics, spatial correlation, and gendered effects.

	Never Married/Single Individuals			Only Heads of Household		
	Direct Effect	Indirect Effect	Total Effect	Direct Effect	Indirect Effect	Total Effect
No controls, no spatial correlation, levels						
Men	N/A	N/A	0.78 **	N/A	N/A	0.83 **
			[0.36]			[0.39]
Women	N/A	N/A	0.73 **	N/A	N/A	0.70 **
			[0.35]			[0.36]
With no controls, only spatial correlation, levels						
Men	1.21 ***	−1.07 *	0.14	1.38 ***	−1.38 **	−0.005
	[0.45]	[0.63]	[0.49]	[0.48]	[0.70]	[0.59]
Women	1.21 ***	−1.08 *	0.13	1.52 ***	−1.75 ***	−0.24
	[0.47]	[0.63]	[0.47]	[0.47]	[0.67]	[0.53]
With controls, spatial correlation, changes						
Men	0.34 ***	3.18 **	3.52 ***	0.16 *	0.69	0.85
	[0.10]	[1.29]	[1.38]	[0.10]	[0.96]	[1.04]
Women	0.09	1.36	1.45	−0.13	−0.97	−1.10
	[0.12]	[1.32]	[1.43]	[0.10]	[0.95]	[1.05]

Source: own calculations with data from [37,39]. *** Significant at 1%, ** Significant at 5%, * Significant at 10%.

**Table 8 ijerph-20-05838-t008:** Excess deaths for Hispanics by age, spatial correlation, and gendered effects.

	Young Adults (18 to 35 Years Old)			Medium Age Adults (36 to 53 Years Old)		
	Direct Effect	Indirect Effect	Total Effect	Direct Effect	Indirect Effect	Total Effect
No controls, no spatial correlation, levels						
Men	N/A	N/A	0.75 **	N/A	N/A	0.81 **
			[0.33]			[0.35]
Women	N/A	N/A	0.72 **	N/A	N/A	0.71 **
			[0.33]			[0.33]
With no controls, only spatial correlation, levels						
Men	1.11 ***	−0.92 *	0.19	1.11 ***	−0.92	0.18
	[0.41]	[0.55]	[0.43]	[0.41]	[0.57]	[0.48]
Women	1.18 ***	−1.03 *	0.14	1.11 ***	−1.08 **	.03
	[0.43]	[0.57]	[0.42]	[0.39]	[0.55]	[0.45]
With controls, spatial correlation, changes						
Men	0.19 ***	1.36	1.55 *	0.03	0.19	0.23
	[0.07]	[0.87]	[0.94]	[0.05]	[0.51]	[0.56]
Women	0.002	−0.05	−0.05	−0.23 ***	−1.93 *	−2.17 **
	[0.06]	[0.55]	[0.60]	[0.09]	[1.05]	[1.13]

Source: own calculations with data from [37,39]. *** Significant at 1%, ** Significant at 5%, * Significant at 10%.

**Table 9 ijerph-20-05838-t009:** Excess deaths for Hispanic women, spatial correlation for different subsamples, and control variables.

	Direct Effect	Indirect Effect	Total Effect	Direct Effect	Indirect Effect	Total Effect	Direct Effect	Indirect Effect	Total Effect
Full Sample				Employed			Citizens		
Occupations	0.003	0.64 **	0.65 **	0.01	0.44	0.45	−0.02	0.20 ***	0.18 ***
	[0.03]	[0.31]	[0.33]	[0.04]	[0.30]	[0.32]	[0.02]	[0.06]	[0.06]
Median income	−0.02	−0.02	−0.04	0.002	−0.17 *	−0.16	−0.01	−0.03	−0.04
	[0.02]	[0.03]	[0.03]	[0.02]	[0.10]	[0.10]	[0.02]	[0.03]	[0.03]
Education	−0.01	−0.07	−0.08	−0.004	−0.08	−0.08	−0.002	−0.05	−0.05
	[0.02]	[0.11]	[0.12]	[0.02]	[0.11]	[0.12]	[0.02]	[0.12]	[0.13]
Townsend index	0.03	0.02	0.05	0.03	−0.11	−0.08	0.03	−0.08	−0.05
	[0.03]	[0.22]	[0.24]	[0.03]	[0.19]	[0.21]	[0.03]	[0.17]	[0.19]
Recently arrived migrants				Health Insurance			Never Married/Single		
Occupations	−0.03	0.24	0.21	0.01	0.45 *	0.46 *	0.04	0.07	0.11
	[0.03]	[0.25]	[0.27]	[0.03]	[0.25]	[0.27]	[0.03]	[0.16]	[0.18]
Median income	−0.05 ***	−0.01	−0.06 ***	−0.02	−0.02	−0.04	−0.02	−0.02	−0.04
	[0.02]	[0.02]	[0.02]	[0.02]	[0.03]	[0.03]	[0.02]	[0.03]	[0.03]
Education	−0.03 *	−0.09	−0.12	−0.01	−0.07	−0.08	−0.004	−0.03	−0.03
	[0.02]	[0.07]	[0.08]	[0.02]	[0.11]	[0.12]	[0.02]	[0.11]	[0.12]
Townsend index	−0.01	0.09	0.08	0.04	−0.04	0.002	0.02	−0.09	−0.07
	[0.02]	[0.13]	[0.14]	[0.03]	[0.17]	[0.18]	[0.02]	[0.14]	[0.15]
Heads of household									
Occupations	−0.01	0.16	0.15						
	[0.03]	[0.20]	[0.22]						
Median income	0.01	−0.20 *	−0.19						
	[0.02]	[0.12]	[0.12]						
Education	−0.01	−0.09	−0.10						
	[0.02]	[0.10]	[0.11]						
Townsend index	0.03	0.03	0.06						
	[0.03]	[0.17]	[0.19]						

Source: own calculations with data from [37,39]. *** Significant at 1%. ** Significant at 5%. * Significant at 10%.

## Data Availability

We will provide datasets analyzed or generated during the study only upon request.

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
