# Peer review of "The Most Vulnerable Hispanic Immigrants in New York City: Structural Racism and Gendered Differences in COVID-19 Deaths"

_ijerph, 2023, doi:10.3390/ijerph20105838_

Round 1

Reviewer 1 Report

Thank you very much for sharing this manuscript with me. I think this topic is very important. I have several minor comments.

1. This is a gender-specific analysis; however, the theory section does not discuss anything about the gendered nature of health disparities in the context of HHP or structural discrimination. I would suggest the authors to revisit this section.

2. Figure 1 is not very clear. Please check this again carefully.

3. There are several grammar/style issues throughout the manuscript. Please carefully check the manuscript again. For example, see like 546.

Author Response

Referee number 1:

  1. This is a gender specific analysis.

Answer: Thank you for this and other very helpful comments. We have expanded our discussion of gender and health inequality in the Introduction (see lines 60-72) and also included a new section on gender and women health’s disparity, under our discussion of the theoretical frameworks guiding our study. (See lines 179-204).

  1. Figure 1 is not very clear.

Answer: We have made Figure 1 clearer.

  1. There are several grammar/style issues…

Answer: We have revised style and grammar and also had another scholar copyedit the paper.

Reviewer 2 Report

The study is relevant and interesting, but there are questions for improvement:

What is the objective of this work?

What is the design of this study? We know the study population but not the inclusion and exclusion criteria

In the sections of “The Hispanic Health Paradox”, “Structural racism and its relation to health outcomes”, “Structural racism and Consequences for the Hispanic population during Covid-19”, and “Spatial concentration of deaths”, the studies are presented and even compared. The methodology should comment on the study design, study population, inclusion and exclusion criteria, data collection techniques, sample, sampling, data analysis technique, etc.

The discussion section is missing.

What are the limitations or biases of this study? What are the future directions of this study?

In the conclusions it is necessary to respond to the objective or objectives in this manuscript, there are comparisons of results that must be in another.

There are very old references from the year 1971, 1986, 2004 and others that must be updated.

The references must be written according to the style established by the journal and this must be reviewed.

Author Response

Referee number 2.

  1. What is the objective of this work?

Answer: We have modified the text, and made the paper’s three objectives more specific and clear, here included:  The first objective is to make an estimation of the mortality differential between the Hispanic and the total national population (see line 199). The second objective is to obtain gendered estimations of the mortality differential (see line 245). The third objective is to analyze the main drivers or the mortality differentials or of the results obtained (see line 250).

  1. What is the design of the study?

Answer: The text was modified, and we have included a clearer explanation our study’s design, one which draws on census data. We estimate the following equations using MLS and Random Panel estimators for spatial data and for different subsamples. The paper achieves the three objectives mentioned in 1 with the following analysis.

(6)

(7)

This is explained in detail between lines 200-253.

  1. The methodology should comment on the study design, study population, inclusion and exclusion criteria, data collection techniques, sample, sampling, data analysis technique, etc.

Answer: For the study design, see our answer to comment #2, above. For the study population, in the case of deaths, it uses census data for New York City as explained in lines 256 to 259.

For our analysis of the PUMA characteristics, the data comes the ACS as explained in lines 275 to 276. We have also explained what “PUMA” stands for (on footnote#2).  The inclusion and exclusion criteria were the adult population of New York City as explained in lines 276 and 277.

The data collection techniques are explained in lines 256 to 259 and in lines 275 to 276 for PUMA characteristics.  Our sampling design for death rates includes the months of November 2020 to June 2021 at the level of PUMA, as explained in lines 257 to 258. In the case of PUMA characteristics, it is the 2019, five-year sample, American Community Survey public data available at IPUMS, as explained in lines 275 and 276. The sampling was the entire number of deaths existing in the New York City Health files as explained in line 256. In the case of the ACS, as explained before, our sample is drawn from the adult population from ACS files, as explained in line 276. To obtain objective three, we use different samples which are explained in detail in section 3.4, see lines 375- 489.

The data analysis technique are OLS and Random Panel estimators for spatial data analysis, as explained in the methodology section, see answer to question #2.

  1. The discussion section is missing.

Answer:  We have modified the paper and created section 4. Discussion. It is found between lines 490 and 521.

  1. What are the limitations or biases of the study? What are the future directions of the study?

Answer: We have modified the conclusion section of the study to include the limitations of the study and potential areas of future research. See lines 546 to 563.

  1. In the conclusions it is necessary to respond to the objective or objectives of the manuscript…

Answer:  We have modified the first part of the conclusion to specify what are the results for our three objectives. See lines 525-545.

  1. There are very old references…

Answer: We have tried to use very recent references, unless they are considered classical studies on the topics as is the case of the Townsend Index (Townsend et al, 1988), or in the Hispanic Health Paradox (Markide and Coreil, 1986; Palloni et al 2004, Lara et al 2005). Note that we have also included more recent (2013; 2017) publications on the HHP.

  1. The references must be written according to the style…

Answer: We have changed the style of the references to the journal’s style.

Reviewer 3 Report

The manuscript seeks to answer the question of why has the Hispanic population in the US suffered higher rates of mortality due to COVID-19 despite the fact that many Hispanic immigrants tend to have better health compared to native-born groups. This is a question worthy of study, so additional cases like this one conducted in New York can add to our knowledge about the different factors and mechanisms that contribute to health disparities.

The authors draw from the concepts of the Hispanic Health Paradox and Structural Racism to construct their hypotheses.

1) The Hispanic Health Paradox argues that the social gradient in health does not apply to Hispanic immigrants, even those in lower-class strata where poorer health outcomes are more pronounced, relative to other ethnic groups. So Hispanics should have a lower mortality rate.

2) Structural racism, in contrast, argues that the combined effects of racial discrimination via housing, policing, employment, health care, etc. prejudice Hispanics negatively in terms of health outcomes, so mortality rates would be expected to be higher. Additionally, a gendered analytical framework is used.

The findings include several comparisons and contrasts with other studies about Hispanic mortality rates and COVID-19, as well as several variables. Without going into the weeds of their findings, the important results suggest that the weathering hypothesis (time spent in the US) as well as citizenship/documentation status matter (pg 401-Line 433/4 and pg 402-Line 435) matter the most. Moreover, the findings suggest that men tend to pay with their lives, whereas women paid by losing their jobs.

A few suggestions for improvement:

1) Briefly mention that Hispanic is not a race but an ethnic group. Within the Hispanic population, there are varied races black, mestizo, and white. Consequently, the literature on Juan Crow as a type of discrimination and/or variant of structural racism may apply better.

2) Methodologically, the authors discuss the use of qualitative interviews. However, the results do not mention any qualitative data, just a few references. Best to remove all references to qualitative data and use it for another publication. Otherwise, better explain the types of mixed methods being used and provide some qualitative data in the findings.

Additionally, I was confused about the types of qualitative data being collected. What is the difference between the interview one line 221  to the type fo interview on line 223 (both on page 5): "...follow-up, in-depth inter-221 view following the survey, conducted long-distance, using a survey and then telephone 222 and computer-structured, face-to-face interviews, given restriction for social distancing 223 during the first year of the pandemic (2020-20210. Besides a survey, we conduct in-depth 224 interviews with..." If the interview data is not relevant to the findings, then just delete it.

3. In terms of the analysis, why wasn't age included as a control variable? Age is one of the most significant risk factors related to COVID-19 mortality. Including this variable could affect the results. One explanation for the Hispanic Health Paradox is that Hispanic immigrants tend to be a younger demographic as well as a healthier group as those who are sick or with other co-morbidities putting them at risk of COVID-19 mortality tend not to migrate.

Also, the finding that "individuals that have less time spent in the US" (line 516, page 16) would imply that they would likely be a younger cohort than those who spent more time since they would likely be older and thus more susceptible to COVID-19-related death.

This could also affect the finding about weathering. The weathering hypothesis--that recurrent stress especially those with fewer resources, more anxieties, etc.--tend to age faster than others. But shouldn't age then be included as a control?

Minor Revision:

1. Make sure PUMA or puma is used consistently throughout the text.

2. Include the key findings in the abstract.

Author Response

Referee number 3.

  1. Briefly mention that Hispanic is not a race but an ethnic group.

      Thanks for this and other valuable recommendations.

Answer: An explanation has been included at the start of the paper, first paragraph, page 1, and also explained in Endnote 1.

  1. Methodologically, the authors discuss the use of qualitative interviews…

Answer: We have modified the paper. We have excluded the qualitative study to focus our analysis only on insights from our quantitative analysis of Census data.

  1. Why wasn’t age included as a control variable?...

Answer: have included this analysis in Table 8, where the role of age is explored in the subsamples. This is discussed in detail in lines 450 to 466. It is found that age matters for both men and women. These results help to build the general result and strengthen our conclusions that structural racism cannot alone explain positive death results for men, but the combination of other variables allows us to obtain evidence for the Hispanic Health Paradox (HHP).

  1. Make sure PUMA or puma is used consistently…

Answer:  We are using PUMA consistently throughout the text. In addition, we have provided at the start of the paper, on endnote #2, an explanation of what PUMA stands for.

  1. Include the key findings in the abstract.

Answer: Yes, we have modified the abstract by including our key findings. See lines 14-22.

Reviewer 4 Report

This is an interesting paper, but difficult to read and comprehend.  The authors mix both methods and discussion into the results section.  These need to be separated into three distinct sections.

Although the author purport that this is a both qualitative and quantitative study, they present few descriptions of the qualitative methods and the data from them really does not appear in the paper.

The results of the paper are suspect because the authors use such overblown language.  They start the paper talking about the "decimation" of the Hispanic population during the pandemic (lines 25 and 39).  This is very much an exaggeration.  The paper needs considerable editing for typos, tense, and other errors.  It is not clear whether the authors are native English speakers, and some if the errors may need reading by such a person to catch them.  Capitalize the acronym PUMA.  Define this for the reader.

Because of the mixture of methods and results, with a little discussion thrown in, it is hard to see what the real purpose of the authors is.  They seem intent on comparing their findings to those of others, but never to making a succinct statement of the importance of these results and their potential meaning for policy.

The tables and figures are very poorly labeled, which adds to the difficulty of a reader comprehending the results.

Author Response

Referee number 4.

  1. This is an interesting paper…authors mix both methods and discussion into the results section…these need to be separated…

 Thank you for this and other significant observations.

Answer: We have modified the paper. Methods are presented in section 2, Methods (see lines 194-253). Section 3 Results, subsection 3.1 Data sources, subsection 3.2 Random panel estimation, subsection 3.3 Estimations for Men and Women, subsection 3.4 Use of different subsamples, subsection 3.5 Using less control variables for women, and Section 4 Discussion of results

  1. Although the author purport this is a both qualitative and quantitative study…

Answer: We have modified the paper by excluding the qualitative study from the paper, we now only concentrate and discuss in our methodology and analysis of findings only discussion from the quantitative or Census data.

The results of the paper are suspect because the authors use such overblown language.

Answer: We have modified the paper. The word decimation was eliminated (see lines 25 to 26). We have also verified the style and grammar.

  1. Because of the mixture of methods…

Answer: We have modified the paper, we have excluded the qualitative study. See our response to question #2.

  1. The tables and figures are very poorly labeled…

Answer: We have modified figures and tables according to the style of the journal.

Much appreciated.  Norma Fuentes-Mayorga and Alfredo Cuecuecha

Round 2

Reviewer 3 Report

approved

Author Response

dear reviewer 3 (second review), thank you for taking the time to review our paper again and for your carefull suggestions.  

We have done a thorough spell check and have also deleted small additional spaces or errors introduced by the changes made by our revisions.

I have also reviewed the citations included in the paper, as suggested by another reviewer and have made sure all citations are included in the reference are cited in the paper.

Thank you for your support and for this opportunity to collaborate with IJERP_MDPI.

Sincerely,

Norma Fuentes-Mayorga

Reviewer 4 Report

The authors of this paper have revised it in line with the reviews received.  They have made substantial changes that increase the scientific merit of the paper.  In particular, they have removed the qualitative data which, while interesting and likely support their quantitative analysis, were not collected in such a way that they can rise to the same level of scientific soundness without considerably more discussion of data collection and analysis methods.  

The revised paper is still quite dense, but it is clearer and the data presented more clearly test the proposed hypotheses.  This paper seems to be unique in its examination of the Hispanic Health Paradox in the context of Covid-19.

The only minor comments are (1) the extensive track changes seem to conceal some typos or problems in spacing, which will no doubt be corrected at the copy editing stage, and (2) the authors should review specifically where they refer to racial groups and include Hispanics.  There seem to be some of these instances lingering in the paper.

Author Response

Dear Reviewer, thank you for taking the time to read one more time our paper and provide such helpful comments and careful insights.  We have 

  1. clarified areas that appear a bit 'dense' in the explanations of findings or those of our results, as well as in the discussion and conclusions.
  2. Have also corrected small typos and spacing introduced while doing the sugested reviews and copyeditin.
  3. Have also made sure to specify that our study is focused on Hispanics when we discuss or review literature or previous research focused on ethnic and racial minorities and made sure to specify how this relates to our study. 

Thank you again for your input and careful review and for this opportunity at collaborate with IJERPH_MDPI.

Sincerely,

Norma Fuentes-Mayorga